# Automatic Diagnosis of Diabetic Retinopathy Stage Focusing Exclusively on Retinal Hemorrhage

**DOI:** 10.3390/medicina58111681

**Published:** 2022-11-20

**Authors:** Yoshihiro Tokuda, Hitoshi Tabuchi, Toshihiko Nagasawa, Mao Tanabe, Hodaka Deguchi, Yuki Yoshizumi, Zaigen Ohara, Hiroshi Takahashi

**Affiliations:** 1Inouye Eye Hospital, 4-3, Kanda-Surugadai, Chiyoda-ku, Tokyo 101-0062, Japan; 2Department of Ophthalmology, Saneikai Tsukazaki Hospital, Himeji 671-1227, Japan; 3Department of Technology and Design Thinking for Medicine, Hiroshima University, Hiroshima 734-8553, Japan; 4Department of Ophthalmology, Nippon Medical School, Bunkyo-ku, Tokyo 113-8603, Japan

**Keywords:** fundus ophthalmoscopy, diabetic retinopathy, retinal hemorrhage, deep learning, deep convolutional neural network

## Abstract

*Background and Objectives*: The present study evaluated the detection of diabetic retinopathy (DR) using an automated fundus camera focusing exclusively on retinal hemorrhage (RH) using a deep convolutional neural network, which is a machine-learning technology. *Materials and Methods*: This investigation was conducted via a prospective and observational study. The study included 89 fundus ophthalmoscopy images. Seventy images passed an image quality review and were graded as showing no apparent DR (*n* = 51), mild nonproliferative DR (NPDR; *n* = 16), moderate NPDR (*n* = 1), severe NPDR (*n* = 1), and proliferative DR (*n* = 1) by three retinal experts according to the International Clinical Diabetic Retinopathy Severity scale. The RH numbers and areas were automatically detected and the results of two tests—the detection of mild-or-worse NPDR and the detection of moderate-or-worse NPDR—were examined. *Results*: The detection of mild-or-worse DR showed a sensitivity of 0.812 (95% confidence interval: 0.680–0.945), specificity of 0.888, and area under the curve (AUC) of 0.884, whereas the detection of moderate-or-worse DR showed a sensitivity of 1.0, specificity of 1.0, and AUC of 1.0. *Conclusions*: Automated diagnosis using artificial intelligence focusing exclusively on RH could be used to diagnose DR requiring ophthalmologist intervention.

## 1. Introduction

Diabetic retinopathy (DR) is a leading cause of visual impairment and blindness. Sabanayagam et al. reported a global annual incidence of DR of 2.2–12.7% and progression of 3.4–12.3% [1].

In Japan, a diabetes survey conducted by the Ministry of Health and Welfare in 1997 estimated that there were 6.9 million patients with diabetes mellitus, including 13.7 million people strongly suspected of having diabetes and 13.7 million people with possible diabetes [2]. Among the patients diagnosed with diabetes, 69.0% of males and 70.8% of females underwent fundus examinations, which are necessary to detect DR, whereas around 30% did not see an ophthalmologist despite having diabetes [3].

Recent image processing technology applying deep learning (DL), a machine-learning algorithm, has attracted attention due to its very high classification performance. The application of this technology to medical images is being actively studied. Image diagnosis has also been reported in the field of ophthalmology [4,5,6,7] and it is expected that this technology will improve access to and the financial sustainability and coverage of DR screening programs [8].

Although DL requires a large amount of computation and data, it has a higher accuracy than other methods of machine learning. However, it is a black box method that is too complex to build a learning model; thus, humans cannot interpret the classification rules, which presents problems in clinical applications.

The earliest signs of DR that appear on the retina are microaneurysms (MAs), which are protruded vessel walls, sometimes occurring due to abnormal blood leakage from retinal vessels. Abnormal swelling of blood vessels causes them to break, leading to retinal hemorrhage (RH). RH is similar to MA but is larger and clearly visible in images of the retina. The present study focused on RH as a means of overcoming the black box problem of artificial intelligence (AI) and investigated the accuracy of its use for the automatic diagnosis of DR.

## 2. Participants and Methods

### 2.1. Participants

The study participants included patients with diabetes who attended the Department of Diabetology, Tsukazaki Hospital (Himeji City, Japan). From 12 May 2021 to 30 July 2021, all participants underwent fundus imaging using a nonmydriatic fundus camera (RetinaStation, Nikon, Japan) in the right eye without a mydriatic drug.

### 2.2. Fundus Photography and AI Diagnosis

A nonmydriatic fundus camera was used in combination with an AI system for DR hemorrhage identification, namely a GPU machine (ELSA GeForce RTX 2070 S.A.C, ELSA Japan, Tokyo, Japan) equipped with our AI program. The GPU machine received the image data (Figure 1) exported from the nonmydriatic fundus camera via USB cable and analyzed the image. The obtained fundus photographs were staged and diagnosed, according to the International Classification of DR, by three retinal specialists using the majority vote method. Cases of poor photography due to poor mydriasis were excluded from the analysis at the discretion of the retinal specialist (T.N.).

### 2.3. AI Model

We constructed a dedicated AI model to identify DR hemorrhagic spots in RetinaStation images. We used the general-purpose qubvel/segmentation_models [9] as a framework, U-net [10] as a model structure, and EfficientNet6 [11] as an encoder. The training data consisted of 50,000 uncompressed small areas (batches) of 256 pix × 256 pix from 300 RetinaStation images (4000 pix × 4000 pix) of DR. The 300 images were annotated by one optometrist under the guidance of a retinopathy specialist. The annotation process was performed using Photoshop (Adobe, San Jose, CA, USA). The image was amplified by randomly repositioning and flipping the image left and right, as well as rotating the image 90 degrees and 30 degrees. The GPU machine used for training was a GTX2060 6GB model and was trained for 100 epochs.

### 2.4. Diagnosis of DR

In the present study, our AI extracted the number of DR hemorrhages as the bleed count and the total number of pixels for DR hemorrhages as the bleed area. These data were used to compare the accuracy of the classification method with mild-or-worse nonproliferative DR (NPDR) and moderate-or-worse NPDR. The Support Vector Machine (SVM) method [12] was used to determine the binary classification boundaries. In this study, we created the SVM model using another RetinaStation fundus image of 87-eye dataset (11 eyes with no apparent DR, 25 eyes with mild NPDR, 32 eyes with moderate NPDR, 11 eyes with severe NPDR, and 8 eyes with PDR). The evaluation indices included sensitivity, specificity, and area under the curve (AUC) via K-fold cross validation (K = 3), and confidence intervals were calculated for each. The threshold was set at the point of the maximum Youden Index. The significance level was set at 0.05.

## 3. Results

Among the 322 patients who visited the Department of DM Medicine at Tsukazaki Hospital, consent to participate in the study was obtained from 293 (91%). Of the 293 participants, 204 (69.6%) were scheduled to undergo ophthalmologic examination within 1 year and 89 (30.4%) were eligible for fundus examination.

A total of 19 of the 89 patients eligible for fundus examination were excluded because the ophthalmologist judged that they could not be photographed or diagnosed from the fundus photographs due to mydriasis or other problems. A final total of 70 fundus images were analyzed for definitive diagnosis by the specialist. DR staging confirmed diagnoses of no apparent DR (NDR; *n* = 51, 76.1%), mild NPDR (*n* = 16, 23.9%), moderate NPDR (*n* = 1, 1.5%), severe NPDR (*n* = 1, 1.5%), and proliferative DR (PDR; *n* = 1, 1.5%). The precision and recall values for each pixel in the segmentation model were 0.504 and 0.558, respectively. The mean (SD) numbers of petechiae were: NDR, 1.18 (1.68) px; mild NPDR, 5.07 (3.86) px; moderate NPDR, 36 (0) px; severe NPDR, 58 (0) px; and PDR, 67 (0) px. The mean (SD) total numbers of pixels were: NDR, 183.44 (400.67) px; mild NPDR, 145.05 (1687.15) px; moderate NPDR, 10,956 (0) px; severe NPDR, 21,278 (0) px; and PDR, 35,237 (0) px.

### Performance of the SVM Model

Scatter plots of the hemorrhage count and area were color-coded based on SVM-estimated stage boundaries in the 87-eye dataset for building the SVM model (Figure 2). The hemorrhage count and area tended to increase as the disease became more severe. Spearman’s correlation coefficient was 0.759. The SVM model did not misdiagnose the severe stage of moderate-or-higher NPDR as the naive or mild retinopathy NPDR stage.

The areas under the curve (AUCs) of this SVM model with upper thresholds of no apparent DR, mild NPDR, moderate NPDR, and severe NPDR were 0.897, 0.955, 0.944, and 0.913, respectively. Figure 3 shows their Receiver Operating Characteristic (ROC) curves, respectively. Among the four thresholds, the classification with Mild NPDR as the threshold showed the highest AUC of 0.955 (SVM means support vector machine).

Figure 4 shows the results of hemorrhage identification using the AI created in this study for the 70 fundus photographs with confirmed staging, as described above. The SVM estimated stage boundaries as color-coded maps. Additionally, in this group, the hemorrhage count and area tended to increase as the disease became more severe. Spearman’s correlation coefficient was 0.740. Furthermore, Table 1 shows the confusion table between the stage inferred by the SVM model and the true stage of confirmed diagnosis for the 70 eyes. The detection of mild-or-worse NPDR showed a sensitivity of 0.812 (95% confidence interval: 0.729–0.895), specificity of 0.888 (95% confidence interval: 0.821–0.955), and AUC of 0.884 (95% confidence interval: 0.775–0.993), and the detection of moderate-or-worse NPDR showed sensitivity, specificity, and AUC values of 1.0 (Figure 5) (we did not set thresholds for moderate and severe NPDR because the number of severe cases in this group was too small). Again, for these 70 eyes, the SVM model did not misdiagnose the severe stage of moderate-or-higher NPDR as the naive or mild retinopathy NPDR stage.

## 4. Discussion

The detection of mild-or-worse NPDR showed a sensitivity of 0.812, specificity of 0.888 and AUC of 0.884, and the detection of moderate-or-worse NPDR showed sensitivity, specificity, and AUC values of 1.0. Screening guidelines typically recommend values of >80% sensitivity and specificity [13]. Therefore, the accuracy was slightly lower when we tried to detect more than mild NPDR. However, mild NPDR does not require ophthalmologic treatment as long as macular function is maintained and medical treatment, such as blood glucose control, is performed [14]. The diagnostic accuracy decreased when we tried to detect mild-or-worse NPDR. This was due to the inclusion of cases in which retinal and choroidal blood vessels were judged to be hemorrhages.

As previously reported, DL for the classification and discrimination of every fundus image requires a large amount of computation and data, but it has higher accuracy than other methods of machine learning. However, the black box nature of DL is a problem in clinical applications, as the learning model is too complex for humans to interpret the classification rules. This is because the diagnoses are not discrete and always contain borderline cases, which contradicts the diagnostic methods used by ophthalmologists, where consistency of the diagnostic process itself is required. The present study used a segmentation method to detect and analyze RHs. Although this method has been shown to have lower classification accuracy than the simple classification and discrimination methods using DL, it is characterized by its ability to obtain classification results in an expressive form that humans can understand clearly, making it easy for ophthalmologists to make a judgment based on AI [15]. In addition, the method used in the present study can be used in actual clinical practice to detect DR after the middle stage. Importantly, differentiating patients into low- and high-risk groups may further improve its cost-effectiveness [16,17].

The present study has several limitations. First, there were only three images showing worse than moderate, which means that it is necessary to increase the number of images. Nineteen (21%) poor-quality images were excluded from the study because it was unclear whether the defects were caused by the opacity of the cornea or lens, or small pupils. Second, RH is not unique to DR, also appearing in age-related macular degeneration, retinal vein occlusions, and retinal vasculitis on fundus images. Furthermore, the characteristic fundus findings of DR include microaneurysms, hemorrhages, exudates, and cotton-wool spots [18]. Our algorithm does not apply to situations where the bleeding point in the ischemic PDR eye is conversely more minor. Third, single-field 45-degree photographs centered at the fovea, compared with seven-field photographs, had a sensitivity of 74–86% and specificity of 92–95% [19,20]. However, this procedure is time-consuming and expensive, and is not appropriate in a screening setting. One advantage of single-field fundus photography is the convenience to patients without vision-threatening retinopathy; moreover, it requires less time and light (only one flash is required), and unlike photography of multiple fields, it does not require mydriasis in the majority of patients. Finally, some studies consider PDR and diabetic macular edema or diabetic maculopathy to be sight-threatening DR (STDR), while some other studies include moderate-to-severe NPDR within the category of STDR [14]. In addition, the presence of macular edema is a positive indicator of diabetes, and the tested algorithms do not provide an output for the presence/absence of macular edema.

## 5. Conclusions

Hemorrhage determination for the staging of DR produces good clinical results. Moreover, the detection of RH may improve interpretability in ophthalmologists and resolve the black box issue associated with AI.

## Figures and Tables

**Figure 1 medicina-58-01681-f001:**
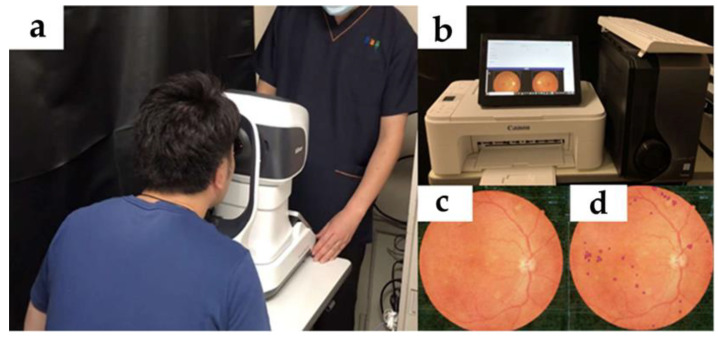
Photographs of the fundus and AI analysis. (**a**) Nonmydriatic fundus camera (RetinaStation, Nikon, Japan). (**b**) Machine used for AI analysis (ELSA GeForce RTX 2070 S.A.C, ELSA Japan, Tokyo, Japan). (**c**) Image of fundus and (**d**) fundus photograph analyzed using AI to identify the areas of hemorrhage, which are marked in red.

**Figure 2 medicina-58-01681-f002:**
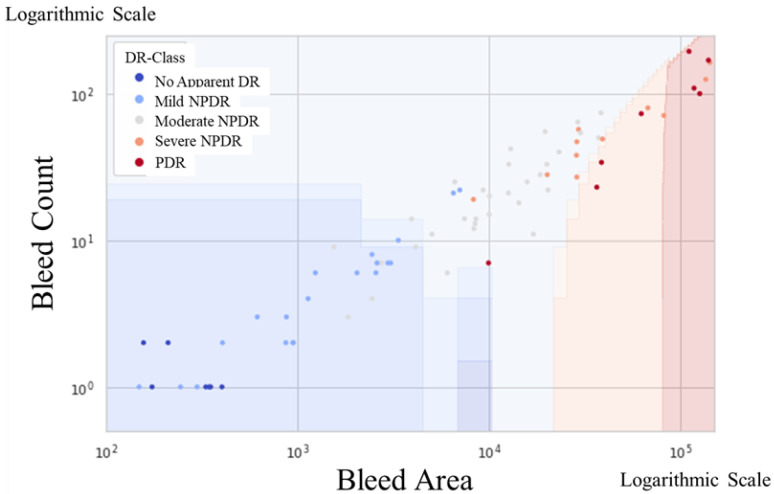
Scatter plots of hemorrhage count and area in the 87-eye dataset for building the SVM model. The SVM model estimated stage boundaries as color-coded maps. The hemorrhage count and area tended to increase as the disease became more severe.

**Figure 3 medicina-58-01681-f003:**
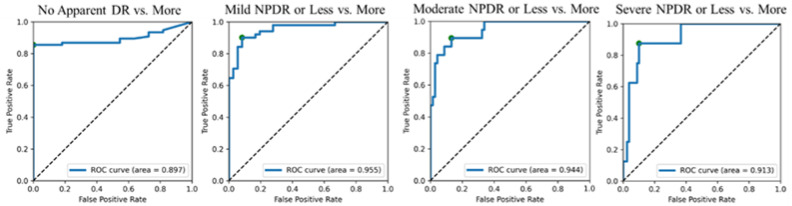
Receiver Operating Characteristic (ROC) curves of four threshold patterns in the SVM classification of the 87-eye dataset for building the SVM model.

**Figure 4 medicina-58-01681-f004:**
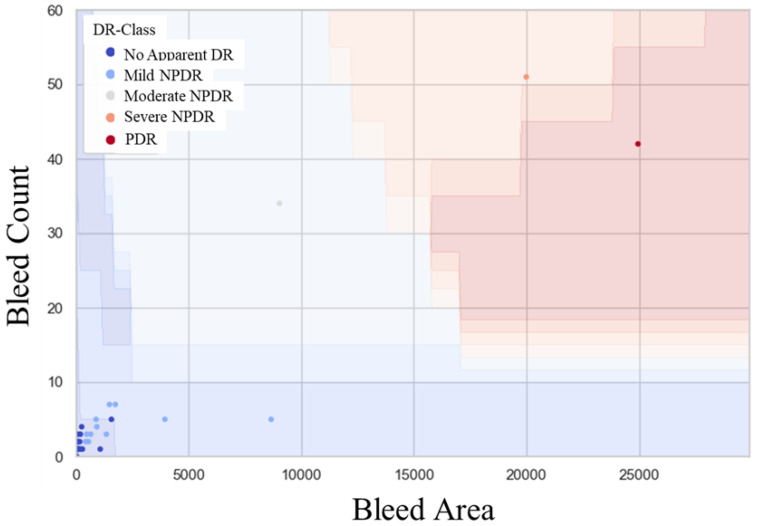
Scatter plots of hemorrhage count and area for eyes of patients attending a diabetes clinic and not scheduled for an eye clinic visit. The SVM model estimated stage boundaries as color-coded maps. The hemorrhage count and area tended to increase as the disease became more severe.

**Figure 5 medicina-58-01681-f005:**
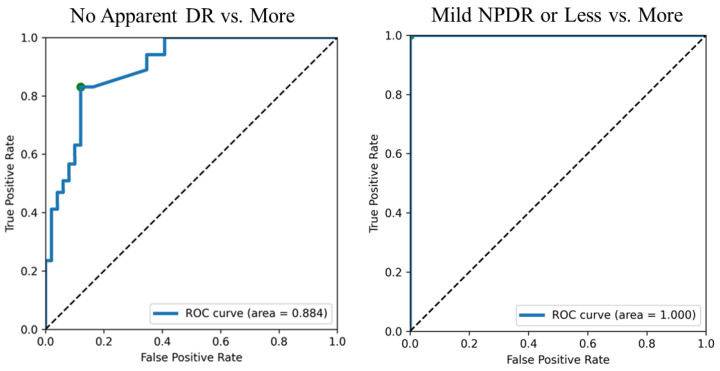
Receiver Operating Characteristic (ROC) curves of two threshold patterns in the SVM classification of 70 eyes of patients attending a diabetes clinic and not scheduled for an eye clinic visit. The classification with mild NPDR as the threshold showed the highest AUC of 1.0 (SVM means support vector machine).

**Table 1 medicina-58-01681-t001:** The confusion-matrix table for 70 eyes of patients attending a diabetes clinic and not scheduled for an eye clinic visit. The rows show numbers of true diagnoses, and the columns show values of SVM prediction. The SVM model did not misdiagnose the severe stage of moderate-or-higher NPDR as the naive or mild retinopathy NPDR stages.

	No Apparent DR	Mild-NPDR	Moderate-NPDR	Severe-NPDR	PDR
No apparent DR	49	5	0	0	0
Mild-NPDR	2	11	0	0	0
Moderate-NPDR	0	0	0	0	0
Severe-NPDR	0	0	0	0	1
PDR	0	0	1	1	0

## Data Availability

Not applicable.

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
