# Peer review of "Automatic Diagnosis of Diabetic Retinopathy Stage Focusing Exclusively on Retinal Hemorrhage"

_medicina, 2022, doi:10.3390/medicina58111681_

Round 1

Reviewer 1 Report (Previous Reviewer 1)

"The earliest signs of DR that appear on the retina are microaneurysms (MAs), which occur due to abnormal blood leakage from retinal vessels" rows 48-49

MAs are not result of abnormal blood leakage, they are protruded vessel walls. In some cases MA can leak, but it is possible to present MA without leakage.

"The obtained fundus photographs were staged and diagnosed according  to the International Classification of Diabetes Mellitus." rows 66-67

My be authors mean Classification of Diabetic Retinopathy?

Author Response

Dear reviewer

Thank you for your point out.

We have changed in file.

Reviewer 2 Report (Previous Reviewer 2)

The suggested corrections have been made. There are no further comments.

Author Response

Dear Reviewer

We appliciated for your accepting our review.

Kind regards,

Toshihiko Nagasawa

This manuscript is a resubmission of an earlier submission. The following is a list of the peer review reports and author responses from that submission.

Round 1

Reviewer 1 Report

Blindness caused by diabetic retinopathy (DR) is well known problem and because of that screening for DR is very important. In nowadays using AI in detection of DR becomes more and more popular. Presented study is actual and interesting. As authors mentioned one very important limitation of the study is very small number of patients/photos with moderate NPDR – one, severe NPDR - one and PDR – one. I am not sure that this numbers are adequate to calculate sensitivity and specificity, and perform statistical analysis. 

Authors used nonmydriatic fundus camera but they wrote “Cases of poor photography due to poor mydriasis were excluded from the analysis …” rows 69-70. It is not clear where they used mydriatics to dilate pupils before photography or no?

On Figure 1 separate pictures are named a,b,d,e , but in text explanation they are a,b,c,d.

Author Response

Thank you for your comments.

I have submitted replay PDF. 

Reviewer 2 Report

The authors present an interesting paper on automatic diagnosis of diabetic retinopathy stage focusing on retinal hemorrhage using AI. 

However, the paper has a serious flaw of very less sample size (moderate, severe NPDR and PDR had only one case each), which makes the results less reliable. Thus, performing ROC in this subgroup doesn't make sense.

Also diabetic retinopathy severity cannot be assessed just with number of hemorrhages. Eyes with extensive ischaemia can have very minimal hemorrhage. 

Author Response

Thank you for your comments. 

I have submitted replya PDF
